# Identification of Stable Reference miRNAs for miRNA Expression Analysis in Adult Neurogenesis Across Mouse and Human Tissues

**DOI:** 10.3390/cells13242060

**Published:** 2024-12-13

**Authors:** Daniella Liana Levitis, Julia Si, Kushal Ravishankar, Michal Toborek, Minseon Park

**Affiliations:** 1College of Art and Science, University of Miami, Coral Gables, FL 33124, USA; dll130@miami.edu (D.L.L.); jxs7123@miami.edu (J.S.); kravis0409@gmail.com (K.R.); 2Department of Biochemistry and Molecular Biology, University of Miami Miller School of Medicine, Miami, FL 33136, USA; mtoborek@med.miami.edu; 3Institute of Physiotherapy and Health Sciences, The Blood-Brain Barrier Research Center, The Jerzy Kukuczka Academy of Physical Education, 40-065 Katowice, Poland

**Keywords:** neurogenesis, neural progenitor cell, hippocampus, miRNA, endogenous control, brain, reference miRNA

## Abstract

Accurate normalization in miRNA studies requires the use of appropriate endogenous controls, which can vary significantly depending on cell types, treatments, and physiological or pathological conditions. This study aimed to identify suitable endogenous miRNA controls for neural progenitor cells (NPCs) and hippocampal tissues, both of which play crucial roles in neurogenesis. Using small RNA sequencing, we identified the most stable miRNAs in primary mouse NPCs and hippocampal tissues and accessed their stability using NormFinder analysis. Six miRNAs—miR-181d-5p, miR-93-5p, miR-103-3p, let-7d-5p, miR-26a-5p, and miR-125a-5p—demonstrated high stability and were evaluated for their suitability as endogenous controls across multiple experimental conditions. All selected miRNAs exhibited consistent expression in the NE-4C mouse cell line but not in ReNcells, a human cell line. For ReNcells, only miR-186-5p, one of the known reference miRNAs tested for comparison, showed stable expression. Notably, miR-103-3p and let-7d-5p were stably expressed in hippocampal tissues from both mouse and human samples but were absent in human brain pericytes, human brain microvascular endothelial cells, and SVG p12 cells, a human fetal glial cell line. This study is the first to identify optimal reference miRNAs for adult neurogenesis in both mouse and human samples, providing reliable options for miRNA normalization and improving the accuracy and reproducibility of miRNA expression analyses in neurogenesis research.

## 1. Introduction

MicroRNAs (miRNAs) are a class of regulatory molecules composed of small non-coding RNAs, typically 21 to 23 nucleotides in length. In most cases, mature miRNAs regulate gene expression post-transcriptionally by interacting with the 3′ untranslated region (UTR) of target mRNA transcripts, marking them for either degradation or recycling prior to translation [1]. miRNAs play a critical role in various biological functions, including determining cell fate by regulating development, differentiation, or apoptosis, and are also associated with human diseases [2,3,4].

Recent studies have demonstrated that miRNAs play a role in regulating neurogenesis [5]. The hippocampal dentate gyrus (DG) and the subventricular zone (SVZ) of the lateral ventricle are niches where neural stem cells (NSCs) and neural progenitor cells (NPCs) reside, continuously proliferating and differentiating to generate new neuronal cells. In the hippocampus, miRNAs regulate processes such as dendritic growth [6] and axonal growth [7], as well as synaptic development, maturation, and plasticity [7]. Moreover, a study showed that miR-7 post-transcriptionally controls *Nlrp3* gene expression, and stereotactic injection of miR-7 mimics into the lateral ventricles significantly inhibited NLRP3 inflammasome activation induced by α-Synuclein while also improving adult neurogenesis in the mouse SVZ [8]. Consequently, abnormalities in miRNA-regulated pathways can lead to impaired neural development, deficiencies in brain connectivity, and embryonic lethality [7,9,10].

Although the role of miRNAs in neurogenesis is widely recognized, the careful selection of endogenous controls for normalizing target miRNA expression has often been overlooked. Endogenous miRNAs can serve as optimal reference genes, as their expression is influenced by the same variables as the target genes. However, since endogenous controls can vary depending on factors such as species, tissue type, or the pathogenic state of the samples, identifying appropriate endogenous controls is essential for accurately investigating the roles of miRNAs in neurogenesis [11].

Several disease models and physical activities are relevant to the study of neurogenesis: (1) substance misuse can impair the proliferation stage of neurogenesis [12]; (2) viral infections are known to inhibit neurogenesis [13,14]; and (3) physical exercise, particularly in the hippocampus, is widely recognized for promoting neurogenesis [15]. To identify endogenous control miRNAs applicable to studies on these specific conditions, we performed miRNA sequencing analysis with primary mouse NPCs isolated from the SVZ of mice exposed to chronic methamphetamine (METH), a highly addictive stimulant that affects the central nervous system (CNS), or infected with ecotropic HIV-1/NDK (EcoHIV), a chimeric HIV-1 virus where the coding region of its surface envelope glycoprotein (gp120) is replaced with the envelope-coding region from ecotropic MLV, restricting the virus’s replication to rodents [16]. Additionally, miRNA sequencing was conducted on dissected hippocampi from voluntarily exercised mice or sedentary control mice.

Based on the analysis, we selected eight stably expressed miRNAs and evaluated their suitability as endogenous controls across various conditions: (1) NE-4C mouse NPCs treated with lipopolysaccharide (LPS) or METH; (2) ReNcell, a human NPC treated with METH and/or HIV-1/NL4-3; (3) mouse hippocampi dissected from mice treated with METH and EcoHIV, followed by either voluntary exercise or sedentary housing; and (4) human hippocampi from control or infarcted brains.

## 2. Materials and Methods

### 2.1. Animal Model

All animal experiments were approved by the University of Miami Institutional Animal Care and Use Committee and were performed in accordance with National Institutes of Health (NIH) guidelines. All animals were housed in an AALAC-accredited facility. Male C57BL/6 mice (13 weeks old; Jackson laboratory, Bar Harbor, ME, USA) were used for the chronic METH abuse and EcoHIV mouse models, which were established in our laboratory as described previously [17,18]. For the voluntary exercise model, both male and female C57BL/6 mice (13 weeks old; Jackson laboratory) were used. Each gender was divided into two groups: a control group housed in locked-wheel cages and an exercise group housed in wheel-equipped cages for two weeks, as described previously [19]. Mice in the exercise group had free access to the running wheel in each cage, while control mice could not use the wheel because it was secured with a cable tie. We also used a combined model that incorporated METH abuse, EcoHIV exposure, and the voluntary exercise model. METH abuse and/or EcoHIV-exposed mice were housed in wheel-equipped or locked-wheel cages for four weeks (Table 1).

To isolate NPCs from the SVZ, the brains were crosscut at bregma +0.2 and ~1 mm, and a thin layer of tissue surrounding the lateral ventricle walls was removed. Isolation of NPCs from the dissected tissues was performed using the NeuroCult™ Enzymatic Dissociation Kit (Stemcell Technology, Vancouver, BC, Canada), with slight modifications as described previously [17].

### 2.2. Small RNA Sequencing and Data Processing

Total RNA was extracted using miRNeasy Mini Kit (Qiagen, Germantown, MD, USA), digested with RNAse-free DNAse I (Epicentre, Madison, WI, USA), filtered through Nanosep 100 K Omega columns (Pall Corporation, Port Washington, NY, USA), and re-purified on RNeasy MinElute columns (Qiagen) using an alternative high-ethanol binding condition to preserve low molecular weight RNA. Then, cDNA libraries were constructed from 100 ng of RNA samples using the CleanTag Small RNA Library Prep Kit (TriLink Biotechnologies, San Diego, CA, USA) following the manufacturer’s instructions. The libraries were quantified using the KAPA Library Quantification Kit (Kapa Biosystems, Wilmington, MA, USA).

The libraries were pooled at equimolar concentrations and diluted prior to loading onto the flow cell of the Illumina NextSeq 500. The libraries were extended and bridge amplified to create sequence clusters and sequenced with 76 nt single-end reads plus 6 nt single-index reads using the NextSeq 500 reagent kit v2 (Illumina, San Diego, CA USA. Cat# FC-404-2005), controlled by the NextSeq Control Software version 2.2.0.4. Real-time image analysis and base calling were performed on the instrument using the Real-Time Analysis (RTA) software version 2.4.11. Base calls from the NextSeq 500 RTA were converted to sequencing reads in FASTQ format using Illumina’s bcl2fastq program v2.17.1.14 with default settings. Only reads passing Illumina’s quality filter were retained for analysis. DESeq2 was used to examine the effect of variables on miRNA expression levels. Library preparation, sequencing, and generating FASTQ files were performed by Frontage Laboratories (Deerfield Beach, FL, USA) [20].

### 2.3. Cell Culture and Treatments

Human brain vascular pericytes (ScienCell Research Lab) were maintained in Pericyte Medium (ScienCell Research Lab). HBMEC Cell Lines (Cell Systems) were maintained in Complete Classic Medium (Cell Systems). ReNcell VM human neural progenitor cell lines (Millipore) were maintained in ReNcell NSC Maintenance Medium (Millipore) supplemented with 20 ng/mL of FGF and 20 ng/mL of EGF. NE-4C (ATCC) was maintained in Eagle’s Minimum Essential Medium (EMEM) (Gibco, Waltham, MA USA) with 10% of fetal bovine serum (FBS; Gibco). All cells were grown in 5% CO_2_ at 37 °C. The treatment conditions for in vitro experiments are described in Table 1. The differentiation of ReNcells was induced by culturing cells without growth factors.

### 2.4. cDNA Synthesis and Real-Time Quantitative PCR (RT-qPCR)

Total RNA, including small RNAs, was extracted using the miRNeasy Mini Kit and Qiashredder (Qiagen) according to the manufacturer’s protocol. RNA concentrations were measured using the NanoDrop 2000 Spectrophotometer (ThermoFisher Scientific, Waltham, MA, USA), and equal concentrations of RNA were used for cDNA production in each experimental cohort to compare the relative expression levels of target miRNAs. cDNA synthesis was performed using the TaqMan Advanced miRNA Assay Kit (ThermoFisher Scientific), and each reaction was carried out with the C1000 Touch Thermal Cycler (Bio-Rad Laboratories, Hercules, CA, USA). The 7500 Real-Time PCR System (ThermoFischer Scientific) with the 7500 Software v2.0.5 was used for qPCR. All samples were run in triplicate, and the average Cq was used. The assay reagents were from ThermoFisher Scientific, and the information is listed in Table 2.

### 2.5. Statistical Analysis

Data are presented by means ± SEM. Except for DESeq2 differential gene expression, statistical analysis was performed using a one-way ANOVA with Tukey’s multiple comparison test, using GraphPad Prism software (version 6.05). Values of *p* < 0.05 were considered statistically significant.

## 3. Results

### 3.1. Selection of Reference miRNA Candidates

To identify the most reliable reference miRNAs for measuring miRNA expression in NPCs, we selected the most stably expressed miRNAs from a pool of 478 mature miRNAs obtained through small RNA sequencing of primary mouse NPCs isolated from the SVZ (SVZ NPCs). The mice were exposed to METH and/or infected with EcoHIV, with a total of 12 mice used, as detailed in Table 1. We measured the reads per million mapped read (RPM) for each miRNA across individual mice, with values ranging from zero to just under 1.5 million. To focus on miRNAs with sufficient expression, we applied a cutoff, excluding those with an average RPM below 5000 across the 12 mice. Next, we calculated the percentage change in standard deviation (SD) relative to the average for each miRNA and ranked them in descending order based on their variability (Appendix A). Stability values for the selected miRNAs, obtained through NormFinder analysis, were all below 0.15, the established cutoff for suitability [21]. Using the same approach, we selected the most stable miRNAs from 698 mature miRNAs expressed in dissected mouse hippocampal tissues, another niche for NSC/NPCs [22] (Appendix A). This cohort included both male and female mice, with some given voluntary running for two weeks and others kept in sedentary housing (Cohort I), as shown in Table 1.

Among the sorted miRNAs, miR-181d-5p and miR-103-3p [23,24] were the most stably expressed in mouse NPCs and hippocampal tissues, respectively. Additionally, miR-26a-5p [25] and miR-125a-5p [26] were stably expressed in both cohorts. To validate their suitability as reference miRNAs across various cell types or human HP tissue, we selected eight miRNA candidates: the top two miRNAs from primary NPCs (miR-181d-5p and miR-93-5p) [27,28,29] and from HP tissues (miR-103-3p and let-7d-5p), the miR-26a-5p ranked third in NPC samples and fifth in HP samples, and the miR-125a-5p ranked fifth in NPC samples and ninth in HP samples. For comparison, we also included miR-186-5p [30,31] and miR-361-5p [32,33], which are used as reference miRNAs across various sample sources (Table 3).

### 3.2. Validation of Selected miRNAs in Mouse and Human NPCs

NE-4C mouse NPCs were treated with either LPS (10 ng/mL) or METH (100 µM) for 2 h. LPS is known to induce inflammation in NE-4C cells [34,35], while METH exposure activates NFκB and SP1, which regulate gene expression [18]. The variability of raw Cq value obtained by RT-qPCR for the selected 8 miRNAs in the NE-4C samples is presented in Figure 1A. All miRNAs were successfully amplified, with mean raw Cq values ranging between 20 and 35. A one-way ANOVA with Tukey’s multiple comparisons test revealed no significant differences between the treated groups and the control group (Figure 1B–I). NormFinder stability values were consistently low, mostly below 0.05 for all miRNAs, with the best combination being miR-26a-5p and miR-361-5p, which had a stability value of 0.002.

To validate the reference miRNA candidates using human NPCs, ReNcells were treated with METH (100 µM) for 24 h or cultured in growth-factor-free media, creating conditions that induce differentiation. Among the candidate miRNAs, miR-103-3p and let-7d-5p were not effectively expressed under the same PCR conditions (undetermined), whereas the remaining miRNAs were well expressed, with mean raw Cq values ranging between 20 and 25 (Figure 2A). NormFinder stability values were consistently low, mostly below 0.05 for all miRNAs, with the best combination being miR-93-5p and miR-361-5p, which had a stability value of 0.006. However, a one-way ANOVA with Tukey’s multiple comparisons test revealed significant differences between the treated groups for most candidate miRNAs, except for miR-186-5p (Figure 2B–G). Based on the results, selected reference miRNAs can be used for NE-4C mouse NPCs, but only miR-186-5p can be used for both mouse and human NPCs.

### 3.3. Validation of Selected miRNAs in Hippocampal Tissues

Hippocampal tissue was dissected from mice that had been exposed to chronic METH and infected with EcoHIV, followed by either voluntary running (Exer) or sedentary housing (Sed) for 4 weeks as previously described [18,36,37] (Cohort II in Table 1). The variability of raw Cq value obtained by RT-qPCR for 8 candidate miRNAs across all samples is presented in Figure 3A. All tested miRNAs were successfully amplified with the mean of raw Cq values ranging between 17 and 25, and NormFinder stability values were very low, mostly below 0.05 for all miRNAs. According to NormFinder, the best combination was miR-186-5p and miR-125a-5p, with a stability value of 0.003. However, a one-way ANOVA with Tukey’s multiple comparisons analysis revealed significant differences between treated groups and the control group for most of the candidate miRNAs, except miR-103-3p and let-7d-5p (Figure 3B–I).

To validate with human tissues, human hippocampal tissue samples were obtained from post-mortem normal brains or stroke-affected brains. The variability of raw Cq value obtained by RT-qPCR for eight candidates across all samples is presented in Figure 4A. All tested miRNAs were successfully amplified, with mean raw Cq values ranging from 21 to 27. NormFinder stability values were very low, all below 0.01, with the most stable combination being miR-93-5p and miR-186-5p, which had a stability value of 0.003. However, a one-way ANOVA with Tukey’s multiple comparisons analysis revealed significant differences between treated groups and control group for miR-93-5p and miR-186-5p (Figure 4C,H), while the other miRNAs showed no significant differences (Figure 4B,D,E,F,G,I). Based on these results, miR-103-3p and let-7d-5p were identified as reliable reference miRNAs in both mouse and human HP tissues. Additionally, miR-181d-5p, miR-26a-5p, miR-125a-5p, and miR-361-5p were found to be reliable reference miRNAs specifically for human HP tissue. Although a downward trend was observed with the stroke variable, it was not statistically significant.

### 3.4. Validation of Selected miRNAs in Different Human Brain Cells

Based on the results, miR-186-5p emerged as the most reliable reference miRNA in mouse and human neural progenitor cell lines, while miR-103-3p and let-7d-5p can be used as references in both mouse and human hippocampal tissues. Next, we evaluated whether these three miRNAs could also serve as a reference miRNA in other brain cell types, including human brain vascular pericytes, HBMECs, and SVG p12, a human fetal glial cell line. To validate their stability, each cell type was prepared in different conditions, as described in Table 1. SVG p12 cells treated with two doses of LPS were used for RT-qPCR. A one-way ANOVA with Tukey’s multiple comparisons test revealed significant differences between the control and the treated groups (Figure 5A–C). Therefore, the selected miRNAs are not suitable for reference miRNAs in human glial cells. In HBMECs, the expression of miR-103a-3p (human ortholog of mouse miR-103-3p) was decreased by HIV infection (Figure 5F), but the expressions of both miR-186-5p and let-7d-5p were unaffected by the treatments (Figure 5D,E). The expression of these two miRNAs was also unchanged in human pericytes treated with LPS or METH (Figure 5G,H), suggesting that miR-186-5p and let-7d-5p can be used as reference miRNAs in HBMECs and pericytes. However, the expression of miR-103a-3p was not amplified (undetermined) in human pericytes under the same conditions of qPCR. 

## 4. Discussion

Alterations in miRNA expression have been closely linked to a wide range of human diseases and developmental processes, leading to extensive research over the past two decades to identify disease-specific target miRNAs and their associated pathways. Accurate quantification of the target miRNA levels is crucial for understanding their significance as biomarkers and their functional roles. Despite the progress, challenges remain in selecting appropriate reference miRNAs for normalization. A recently published study used small non-coding RNAs (snRNA) such as RNU6 and RNU5G, or small nucleolar RNA (snoRNA) SNORD48, as endogenous controls to identify a specific miRNA as a therapeutic target for the treatment of Alzheimer’s disease (AD) [38]. However, other studies suggest that using the same class of RNAs for normalization is preferable due to their shared properties, such as RNA stability and size [39]. Additionally, since miRNA content constitutes as little as 0.01% of the total RNA mass, with concentrations varying greatly by sample type [40], the chosen endogenous control miRNA should reflect these quantitative changes in the target miRNA in the same sample type.

Adult neurogenesis, extensively studied in the hippocampus and SVZ, is a process in which the stages of neuronal development are well characterized [41]. Evidence suggests that NSCs/NPCs originating from the SVZ migrate to brain regions such as the striatum, amygdala, and cortex, a process influenced by stroke, ischemia, and neurodegenerative disorders [42,43,44]. Also, altered hippocampal neurogenesis has also been observed in both rodents and humans with AD [45,46]. Therefore, therapeutic strategies for AD, neurogenesis deficits, and stroke/ischemia increasingly leverage miRNAs due to their ability to regulate multiple targets across diverse cell types [47,48,49]. To investigate the roles of differentially expressed miRNAs in these conditions, selecting appropriate endogenous control miRNAs is essential. As mentioned earlier, since miRNA expression varies depending on species, tissue type, and disease pathology, identifying suitable reference miRNAs for SVZ NPCs and hippocampal tissues is crucial to understanding miRNAs’ role in neurogenesis within these regions.

To identify the most suitable reference miRNAs for neurogenesis in the SVZ and hippocampus under various conditions, we extracted small RNA-enriched total RNAs from primary ex vivo cultures of SVZ NPCs and dissected hippocampi of mice subjected to different treatments. These treatments included chronic METH exposure, EcoHIV infection, METH, EcoHIV comorbidity, and voluntary exercise in male and female mice, with wild-type male and female mice included as controls (Table 1). A total of 478 mature miRNAs were identified in the mouse SVZ NPCs, while 698 mature miRNAs were identified in the mouse hippocampal tissue. Although NSCs/NPCs are present in the hippocampal DG, the primary cell types in the mouse hippocampus include excitatory neurons, inhibitory interneurons, astrocytes, microglial, oligodendrocytes, and cerebral endothelial cells [50]. Consequently, the miRNA pool from the hippocampal tissue may not reflect miRNA profiles specific to the hippocampal NSCs/NPCs. However, when we compared abundantly and stably expressed miRNAs from both datasets, miR-26a-5p, the third most stably expressed miRNA in SVZ NPCs (Appendix A), was the fifth most stable miRNA in the hippocampal samples (Appendix A). Similarly, let-7d-5p ranked 2nd in the hippocampal list and 15th in the SVZ NPCs list. Although not included in the selected candidates for validation, miR-125a-5p appeared among the top 15 in both datasets, ranking 5th in the SVZ NPCs and 11th in the hippocampal samples. These findings highlight a subset of miRNAs with stable expression across distinct neural regions and conditions, suggesting their potential as reliable reference miRNAs for neurogenesis studies in the SVZ and hippocampus.

When validating the selected reference miRNA candidates in independently prepared samples treated with different stimuli, we observed discrepancies between NormFinder and one-way ANOVA analyses. For instance, NormFinder identified miR-93-5p as the most stable miRNA, with a combination of miR-93-5p and miR-361-5p recommended as optimal multiple controls in ReNcells. However, one-way ANOVA revealed a statistically significant difference in the expression of both miRNAs between the METH-treated and proliferating-conditioned groups (Figure 2C,G). NormFinder evaluates overall stability by minimizing both intergroup and intragroup variation, ensuring consistent expression across all samples without considering specific treatment or condition differences [21]. In contrast, one-way ANOVA assesses statistical differences in mean expression between groups, making it sensitive to minor shifts in mean expression, particularly when intragroup variation is low. Consequently, while NormFinder may identify miRNAs as stable across conditions, ANOVA can reveal specific intergroup differences, occasionally resulting in seemingly contradictory results. NormFinder identified the best reference miRNA and the best combination of two miRNAs. In some cases, the stability of the combination was lower than that of the individual miRNAs. For example, in the human HP samples, the most stable combination was miR-93-5p and miR-186-5p, with a stability value of 0.003. However, their individual stability values were 0.004 and 0.013, respectively. NormFinder calculates stability based on intra-group and inter-group variations, where group variance is weighted by the number of samples in each group. These two variations are then combined into a single stability value. This process explains why the stability of a gene combination can differ from the stability of individual miRNAs [21].

Our study is the first to identify the most optimal reference miRNAs for miRNA research in adult neurogenesis, with specific applicability to NSC/NPCs, hippocampal tissue, HBMECs, pericytes, and glial cells, as summarized in Table 4. All selected miRNA candidates showed stable expressions in a mouse NPC cell line (NE-4C), whereas only miR-186-5p was applicable to the human NPC cell line (ReNcells). For hippocampal tissue samples, miR-103-3p (miR-103a-3p, the human ortholog) and let-7d-5p are the most appropriate reference miRNA. Interestingly, miR-103-3p was either not amplified in human brain pericytes or showed significant expression changes under treatment conditions in HBMEC and SVG p12 cells. This suggests that including miR-103-3p as one of the multiple reference miRNAs would be ideal for accurately analyzing miRNA expression profiles in hippocampal tissues from both mouse and human samples.

## 5. Conclusions

This study provides a comprehensive evaluation of candidate reference miRNAs for miRNA expression analysis in adult neurogenesis across diverse experimental conditions and species. Our findings highlight miR-103-3p (human ortholog: miR-103a-3p) and let-7d-5p as robust and stable reference miRNAs for normalization in hippocampal tissues from both mouse and human samples. However, miR-103-3p may be a more specific reference miRNA for normalizing hippocampal tissues. Additionally, miR-26a-5p and miR-125a-5p demonstrated high stability in various contexts, including NE-4C mouse NPCs, human hippocampal tissues, human pericytes, and HBMECs, making them suitable for targeted applications. Furthermore, miR-186-5p was identified as the most stable reference miRNA for ReNcells, a human NPC line. These results emphasize the critical importance of selecting appropriate reference miRNAs tailored to specific tissue types, species, and experimental conditions to ensure accurate quantification and interpretation of miRNA expression data. The identified reference miRNAs will support future research into miRNA-regulated processes in neurogenesis, stroke, ischemia, and neurodegenerative disorders, contributing to a deeper understanding of their roles in central nervous system function and pathology.

## Figures and Tables

**Figure 1 cells-13-02060-f001:**
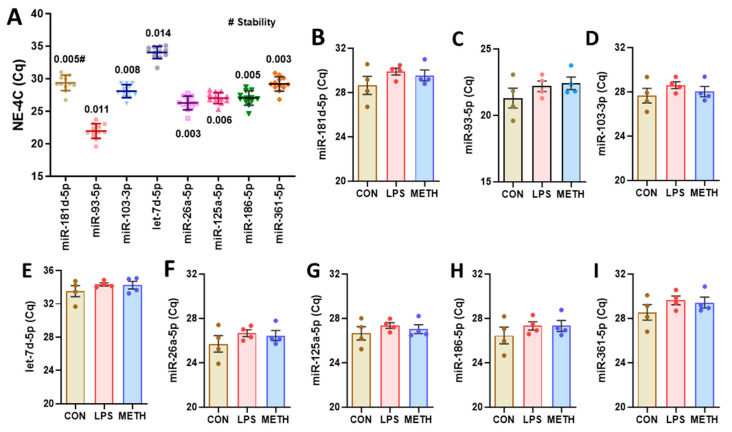
Overview of Cq values obtained by RT-qPCR for NE-4C samples: (**A**) NE-4C mouse NPCs were treated with LPS (10 ng/ml), METH (100 µM), or PBS as a vehicle control for 2 h. Mean ± SD, N = 12 (4 per group). Dots shown in A represent mean raw Cq values for technical replicates of individual samples. Cq values for the indicated miRNAs from all groups were used for NormFinder stability analysis (#). (**B**–**I**) Group comparison of miRNA candidates for NE-4C samples. Data are presented as mean ± SEM, N = 4 per group.

**Figure 2 cells-13-02060-f002:**
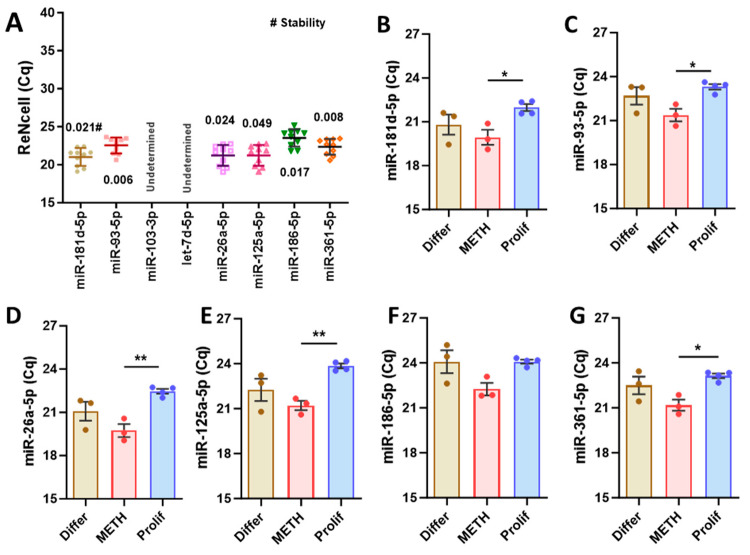
Overview of Cq values obtained by RT-qPCR for ReNcell samples: (**A**) Human NPCs (ReNcells) were treated with METH (100 µM), cultured in growth-factor-free media (a differentiation condition), or cultured in complete media (containing EGF and FGF, a proliferation condition) for 24 h. Data are presented as mean ± SD, N = 11–12 (3–4 per group). Dots represent mean raw Cq values for technical replicates of individual samples. Cq values for the indicated miRNAs from all groups were used for NormFinder stability analysis (#). (**B**–**G**) Group comparison of miRNA candidates for ReNcell samples. Data are presented as mean ± SEM, N = 3–4 per group. * *p* < 0.05, ** *p* < 0.01.

**Figure 3 cells-13-02060-f003:**
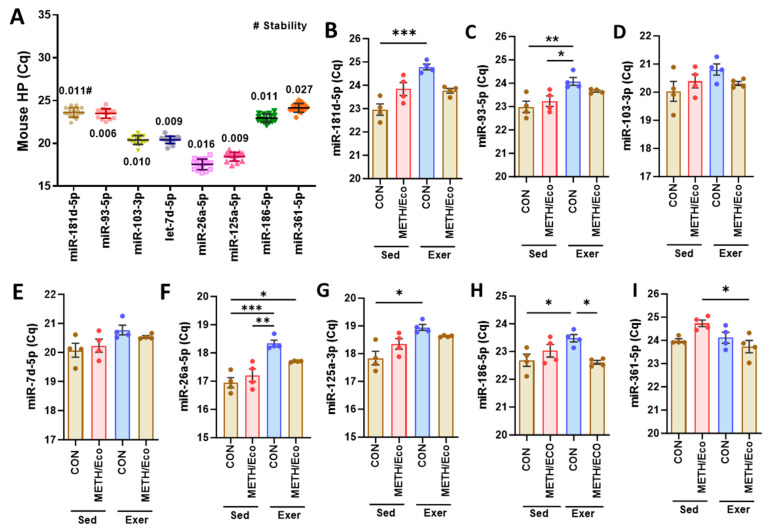
Overview of Cq values obtained by RT-qPCR for mouse hippocampal tissue samples: (**A**) C57BL/6 mice were exposed to chronic METH and infected with EcoHIV, followed by housing in wheel cages for voluntary running for 4 weeks. Total RNAs, including small RNA, were extracted from the dissected hippocampal tissues. Dots represent mean raw Cq values for technical replicates of individual samples. Cq values for the indicated miRNAs from all groups were used for NormFinder stability analysis (#). Data are presented as mean ± SD, N = 4 per group. (**B**–**I**) Group comparisons of miRNA candidates for mouse HP tissues. Data are presented as mean ± SEM, * *p* < 0.05, ** *p* < 0.01, *** *p* < 0.005.

**Figure 4 cells-13-02060-f004:**
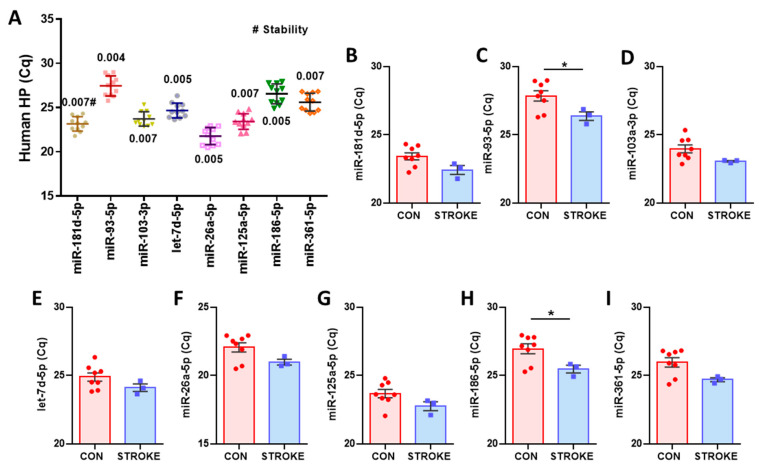
Overview of Cq values obtained by RT-qPCR for human hippocampal tissue samples. (**A**) Total RNA, including small RNA, was extracted from post-mortem human hippocampal tissues from normal brains (N = 7) and stroke-affected brains (N = 3), obtained from the UM Brain Endowment Bank. Dots represent mean raw Cq values for technical replicates of individual samples. Cq values for the indicated miRNAs from all groups were used for NormFinder stability analysis (#). Data are presented as mean ± SD. (**B**–**I**) Group comparison of miRNA candidates for human HP tissues. Data are presented as mean ± SEM, * *p* < 0.05.

**Figure 5 cells-13-02060-f005:**
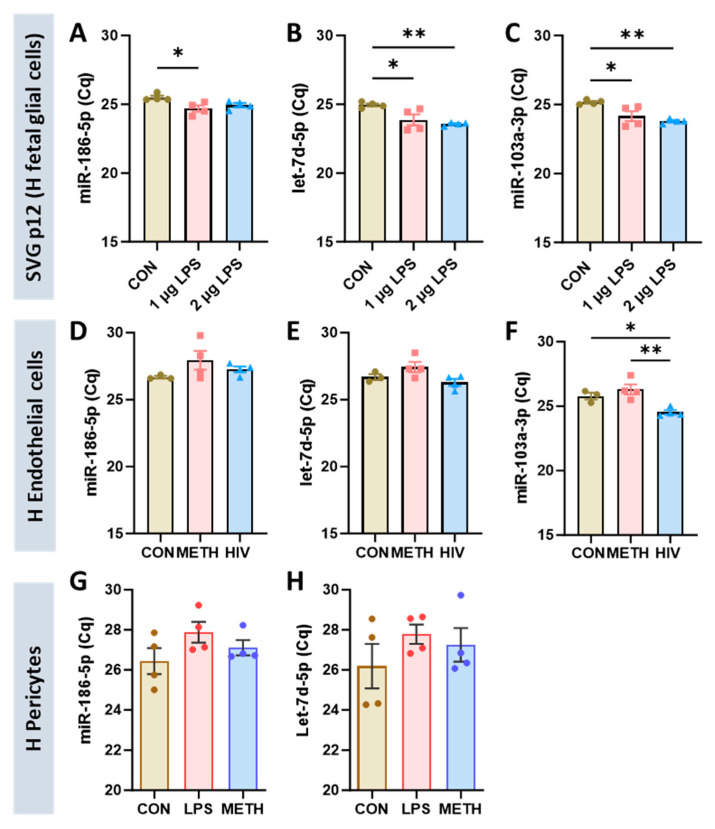
Overview of Cq values obtained by RT-qPCR for SVG p12 cells, HBMECs, and human pericytes: (**A**–**C**). Total RNA, including small RNA, was extracted from SVG p12 cells treated with LPS (1 or 2 µg/mL) for 2 h. A one-way ANOVA with Tukey’s multiple comparisons test was performed. N = 4 per group (mean ± SEM), * *p* < 0.05 and ** *p* < 0.01. (**D**–**F**) Total RNA was extracted from HBMEC-treated METH (100 µM) or HIV NL4-3 (60 ng/ml of p24) for 24 h. N = 4 per group (mean ± SEM), * *p* < 0.05 and ** *p* < 0.01. (**G**,**H**) Total RNA was extracted from human brain vascular pericytes treated METH (100 µM) or LPS (4 µg/mL) for 2 h. miR-103a-3p was not amplified under the same qPCR conditions. N = 4 per group (mean ± SEM).

**Table 1 cells-13-02060-t001:** List of samples used for RT-qPCR.

Sample Name	Organism	Sample Type (Diases)	Source
Human brain vascular pericytes	Human	Primary brain vascular pericytes	ScienCell Research Lab (Carlsbad, CA, USA. Cat#1200)
Pericyte-CON			
Pericyte-METH: Pericyte treated with METH (100 µM) for 2 h			
Pericyte-LPS: Pericyte treated with LPS (4 µg/ml) for 2 h			
HBMEC	Human	Human Brain Microvascular Endothelial Cells	Cell Systems (Kirkland, WA, USA. Cat# ACBRI 376)
HBMEC-CON			
HBMEC-METH: HBMEC treated with METH (100 µM) for 24 h			
HBMEC-HIV: HBMEC infected with HIV (NL 4-3) for 24 h			
SVG p12	Human	Human Fetal Glial cells	ATCC (Manassas, VA, USA. Cat# CRL-8621)
SVG p12- CON			
SVG p12- LPS: SVG p12 treated with LPS (1 or 2 µg/mL) for 2 h			
ReNcell VM cell line	Human	Neural progenitor cells	Millipore (Burlington, MA, USA. Cat# SC008)
ReNcell-METH: ReNcell treated with METH (100 µM) for 24 h			
ReNcell-Differ: ReNcell induced differentiation for 24 h			
ReNcell-Prolif: ReNcell in proliferation condition			
Human HP tissue from post-mortem	Human	Hippocampal tissue	University of Miami’s Brain Endowment Bank™ (Miami, FL, USA)
HP-CON: unaffected control			
HP-stroke: hypoxic ischemic encephalopathy or cerebral infarction			
NE-4C cell line	Mouse	Neural stem cells	ATCC (Cat# CRL-2925)
NE-4C-METH: NE-4C treated with METH (100 µM) for 2 h			
NE-4C-LPS: NE-4C treated with LPS (10 ng/mL) for 2 h			
NE-4C-CON: NE-4C vehicle control			
Primary SVZ NPCs isolated from C57/BL6 mouse brain	Mouse	Neural progenitor cells	Park et al. [17]
NPC-CON: Male mice were injected with saline as a vehicle			
NPC-METH: Male mice exposed to METH			
NPC-EcoHIV: Male mice exposed to EcoHIV			
NPC-METH/EcoHIV: Male mice exposed to METH and EcoHIV			
HP tissue dissected from C57/BL6 mouse brain-Cohort I	Mouse	Hippocampal tissue	Wolff et al. [19]
HP-F-Sed-Con: Female sedentary control mice			
HP-F-Ex-Con: Female mice voluntarily exercised for 2 weeks			
HP-M-Sed-Con: Male sedentary control mice			
HP-M-Ex-Con: Male mice voluntarily exercised for 2 weeks			
HP tissue dissected from C57/BL6 mouse brain-Cohort II	Mouse	Hippocampal tissue	
HP-M-Sed-Con: Male mice exposed to mock controls and sedentary housed for 4 weeks			
HP-M-Ex-Con: Male mice exposed to mock controls followed by voluntarily exercised for 4 weeks			
HP-M-Sed-METH/EcoHIV: Male mice exposed to METH and EcoHIV and sedentary housed for 4 weeks			
HP-M-Ex-METH/EcoHIV: Male mice exposed to METH and EcoHIV followed by voluntarily exercised for 4 weeks			

**Table 2 cells-13-02060-t002:** List of TaqMan assays for target miRNAs.

Target miRNA	Target Sequence	Product Name
hsa-miR-93-5p	CAAAGUGCUGUUCGUGCAGGUAG	478210_mir
hsa-miR-186-5p	CAAAGAAUUCUCCUUUUGGGCU	477940_mir
hsa-miR-361-5p	UUAUCAGAAUCUCCAGGGGUAC	478056_mir
hsa-miR-181d-5p	AACAUUCAUUGUUGUCGGUGGGU	479517_mir
hsa-let-7d-5p	AGAGGUAGUAGGUUGCAUAGUU	478439_mir
hsa-miR-26a-5p	UUCAAGUAAUCCAGGAUAGGCU	477995_mir
hsa-miR-125a-5p	UCCCUGAGACCCUUUAACCUGUGA	477884_mir
hsa-miR-103a-3p	AGCAGCAUUGUACAGGGCUAUGA	478253_mir
mmu-miR-93-5p	CAAAGUGCUGUUCGUGCAGGUAG	mmu481280_mir
mmu-miR-186-5p	CAAAGAAUUCUCCUUUUGGGCU	mmu_480966_mir
mmu-miR-361-5p	UUAUCAGAAUCUCCAGGGGUAC	mmu_481127_mir
mmu-miR-181d-5p	AACAUUCAUUGUUGUCGGUGGGU	mmu_479517_mir
mmu-let-7d-5p	AGAGGUAGUAGGUUGCAUAGUU	mmu478439_mir
mmu-miR-26a-5p	UUCAAGUAAUCCAGGAUAGGCU	mmu_481013_mir
mmu-miR-125a-5p	UCCCUGAGACCCUUUAACCUGUGA	mmu_480906_mir
mmu-miR-103-3p	AGCAGCAUUGUACAGGGCUAUGA	mmu_478253_mir

**Table 3 cells-13-02060-t003:** Selected endogenous miRNA candidates from primary cultured mouse SVZ NPCs and mouse hippocampal tissues.

Name of Mature miRNA	Sample Sources	Average RPM *	SD	SD/Avg (%)	Stability **	References
mmu-miR-181d-5p	mouse SVZ NPCs	17,133.43	941.94	5.50	0.037	N/A ^a^
mmu-miR-93-5p	mouse SVZ NPCs	12,526.65	799.50	6.38	0.072	[27,28,29]
mmu-miR-103-3p	mouse HP tissues	7957.05	316.53	3.98	0.041	[23,24]
mmu-let-7d-5p	mouse HP tissues	12,732.27	539.82	4.24	0.046	
mmu-miR-26a-5p	mouse SVZ NPCs	234,194.56	16,500.17	7.05	0.079	[25]
	mouse HP tissues	40,821.93	2284.65	5.60	0.041	
mmu-miR-125a-5p	mouse SVZ NPCs	26,477.70	2133.54	8.06	0.061	[26]
	mouse HP tissues	9999.43	705.37	7.05	0.057	

* RPM: reads per million mapped read. ** stability: Reference gene stability value by NormFinder; a lower value indicates greater stability. ^a^ N/A: Not available (PubMed. Google Scholar, or Web of Science).

**Table 4 cells-13-02060-t004:** Summary of qPCR validation results for selected miRNA candidates across sample types.

Souce of Selected miRNA	Reference miRNA Candidate	NE-4C Cell Line (M): con, LPS. METH (N = 4)	ReNcell VM Cell line(H):Prolif, Diff, METH (N = 3~4)	Mouse HP Tissues: con-Sed, MH-Sed, con-Exer, MH-Exer (N = 4)	Human HP Tissues: Control HP (N = 8), Stroke HP (N = 3)	H Pericytes: con, LPS, METH (N = 4)	HBMEC: con, METH, HIV (N = 3~4)	SVG p12 (H Fetal Glial Cells):con, LPS (N = 4)
Mouse SVZ NPCs	miR-181d-5p	Stable	* *p* < 0.05	*** *p* < 0.005	Stable	Stable	** *p* < 0.01	** *p* < 0.01
	miR-93-5p	Stable	* *p* < 0.05	** *p* < 0.01	* *p* < 0.05	Stable	Stable	** *p* < 0.01
Mouse HP tissues	miR-103-3p	Stable	UD ^a^	Stable	Stable	UD	** *p* < 0.01	** *p* < 0.01
	let-7d-5p	Stable	UD	Stable	Stable	Stable	Stable	** *p* < 0.01
Both in mouse NPCs and HP	miR-26a-5p	Stable	** *p* < 0.01	*** *p* < 0.005	Stable	Stable	Stable	** *p* < 0.01
	miR-125a-5p	Stable	** *p* < 0.01	* *p* < 0.05	Stable	Stable	Stable	Stable
Known Control miRNAs	miR-186-5p	Stable	Stable	* *p* < 0.05	* *p* < 0.05	Stable	Stable	* *p* < 0.05
	miR-361-5p	Stable	* *p* < 0.05	* *p* < 0.05	Stable	Stable	Stable	*** *p* < 0.005

^a^ UD: undetermined.

## Data Availability

Data are contained within the article and Appendix A.

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
