# Peer review of "Identification of Stable Reference miRNAs for miRNA Expression Analysis in Adult Neurogenesis Across Mouse and Human Tissues"

_cells, 2024, doi:10.3390/cells13242060_

Round 1

Reviewer 1 Report

Comments and Suggestions for Authors

The study entitled “Identification of Stable Reference miRNAs for miRNA Expression Analysis in Adult Neurogenesis Across Mouse and Human Tissues” by Daniella Levitis et al. used NormFinder analysis to determine the stability of the most stable miRNAs found in primary mouse NPCs and hippocampus tissues by small RNA sequencing. When tested under various experimental settings, six miRNAs—miR-181d-5p, miR-93-5p, miR-103-3p, let-7d-5p, miR-26a-5p, and miR-125a-5p showed excellent stability and were judged to be suitable endogenous controls.

According to the authors, this study is the first to pinpoint the best reference miRNAs for adult neurogenesis in both human and mouse samples. This enhances the precision and repeatability of miRNA expression assessments in neurogenesis research and offers dependable choices for miRNA normalization.

The experiments are sufficient and were carried out properly, and the text is neatly written.
The following issues must be resolved before the text can be approved:
1. The table 1 that is referenced in line 154 is absent.
2. The text makes no reference to Tables 2 and 3 on page 4.
3. Since there are no J and K in the figure, (Figure 2B-K) at line 210 should be changed to (Figure 2B-I).
4-Every section of figure 4 needs to be numbered according to lines 234, 237, and 239 in the text.
5. The manuscript needs to have a conclusion section.

Author Response

Dear reviewer,

We sincerely appreciate your kind words regarding our research and your careful comments, which have greatly contributed to improving the completeness of this manuscript. As per your advice, we have made the necessary corrections to the text and figures. Below are the details of these corrections:

  1. The table 1 that is referenced in line 154 is absent.

: We found that Table 1 is missing in the preprint version for some reason. However, it is included in the original Word version. Please verify the presence of Table 1 in the Word file on page 3. Please verify its presence there.

  1. The text makes no reference to Tables 2 and 3 on page 4.

: We have highlighted the lines referencing Tables 2 and 3 on pages 4 and 5, respectively, in the Word file.

  1. Since there are no J and K in the figure, (Figure 2B-K) at line 210 should be changed to (Figure 2B-I).

: This was our mistake, which occurred while preparing multiple versions of the figures. Thank you for pointing this out. We have corrected it at line 213 on page 8 and highlighted the changes. Additionally, the previous Figure 2 has been renumbered to Figure 3 in the revised Word file. As requested by the other reviewer, we have included the qPCR results for individual miRNAs in NE-4C cells as Figure 1.

4-Every section of Figure 4 needs to be numbered according to lines 234, 237, and 239 in the text.

: This oversight occurred during the transfer of the figure from the PowerPoint file to the Word file. We have corrected this error in Figure 5 (formerly Figure 4) on page 9.

  1. The manuscript needs to have a conclusion section.

: As recommended, we have added a conclusion section following the discussion section.

Reviewer 2 Report

Comments and Suggestions for Authors

The authors aimed to identify stable reference miRNAs for miRNA expression analysis in adult neurogenesis across mouse and human tissues. The manuscript addresses an important topic, as selecting appropriate control miRNAs is crucial for accurately evaluating miRNA expression, which can vary significantly depending on species, tissue type, and treatment conditions.

Issues Identified:

  1. Table 1 Missing:
    Table 1 is not included in the manuscript, which hinders the reader's ability to fully understand the context of the data.

  2. Gender-Specific Results:
    On line 88, the authors state, “Each gender was divided into two groups: a control group housed in locked-wheel cages and an exercise group housed in wheel-equipped cages for two weeks.” Later, on line 273, they mention, “These treatments included chronic METH exposure, EcoHIV infection, METH and EcoHIV comorbidity, voluntary exercise in male and female mice, with wild-type male and female mice included as controls (Table 1).”
    However, the results are not presented separately for male and female subjects. Given the existence of sex-biased miRNA expression, it would be beneficial to display the results separately for each gender rather than combining them.

  3. NormFinder Analysis Methodology:
    The manuscript lacks details on how the NormFinder analysis was performed, raising several questions:

    • Figure 1A:
      The authors state, “The best combination is miR-26a-5p and miR-361-5p, with a stability value of 0.002.” However, the individual stability values for miR-26a-5p and miR-361-5p are both 0.003. Clarification is needed: Is 0.002 the combined stability value?

    • Line 183:
      “A one-way ANOVA with Tukey’s multiple comparisons test revealed no significant differences between the treated groups and the control group.” This result is not shown in the figure. Including this data would enhance the manuscript's transparency.

    • Figure 1B:
      The best combination is reported as miR-93-5p and miR-361-5p, with a stability value of 0.006, but the figure indicates individual stability values of 0.006 and 0.008, respectively.

    • Figure 2A:
      The authors claim that miR-186-5p and miR-125a-5p form the best combination with a stability value of 0.003, yet the figure shows individual values of 0.011 and 0.009.

    • Figure 3A:
      The reported combined stability for miR-93-5p and miR-186-5p is 0.003, while the figure shows individual values of 0.004 and 0.005.

  4. Reliability of Reference miRNAs:
    Line 221 states, “miR-181d-5p, miR-26a-5p, miR-125a-5p, and miR-361-5p were the most reliable reference miRNAs in human HP tissue.” The rationale behind this conclusion is unclear. miR-103-3p and let-7d-5p also appear to be reliable reference miRNAs in both mouse and human HP tissues, with let-7d-5p showing the lowest stability value (0.005).

  5. Figure 4:
    Panels A to I are not labeled, which makes it difficult to interpret the data.

Summary:

The data suggest that no single miRNA is universally suitable as a reference across all conditions tested in human or mouse samples. Presenting gender-specific results, clarifying the NormFinder methodology, and ensuring that all data is accurately represented in figures would significantly strengthen the manuscript.

Author Response

Dear reviewer,

We sincerely appreciate your effort for reviewing our manuscript and your careful comments, which have greatly contributed to improving the completeness of this manuscript. As per your advice, we have made the necessary corrections to the text and figures. Below are the details of these corrections:

  1. Table 1 Missing:
    Table 1 is not included in the manuscript, which hinders the reader's ability to fully understand the context of the data.

We found that Table 1 is missing in the preprint version for some reason. However, it is included in the original Word version. Please verify the presence of Table 1 in the Word file on page 3. Please verify its presence there.

  1. Gender-Specific Results:
    On line 88, the authors state, “Each gender was divided into two groups: a control group housed in locked-wheel cages and an exercise group housed in wheel-equipped cages for two weeks.” Later, on line 273, they mention, “These treatments included chronic METH exposure, EcoHIV infection, METH and EcoHIV comorbidity, voluntary exercise in male and female mice, with wild-type male and female mice included as controls (Table 1).”
    However, the results are not presented separately for male and female subjects. Given the existence of sex-biased miRNA expression, it would be beneficial to display the results separately for each gender rather than combining them.

: We agree with the reviewer’s point about the importance of sex-specific reference miRNAs. However, our analysis combined sequencing data from male and female mice to identify common reference miRNAs applicable across sexes. Accordingly, we also used human hippocampal tissues from both male and female subjects for validation, as shown in Figure 4 in the revised Word file.

  1. NormFinder Analysis Methodology:
    The manuscript lacks details on how the NormFinder analysis was performed, raising several questions regarding the difference between single miRNA stability and combination stability in the formerly Figure 1A, Figure 1B, Figure 2A, and Figure 3A.

: I apologize for the confusion caused by the difference between the stability of individual miRNAs and the stability of a combination of genes. NormFinder calculates stability based on intra-group and inter-group variations. It computes the average expression value of the genes for each sample, calculates the standard deviation (SD) of these average values within each group (intra-group variation), and determines the group means of the average expression values. It then calculates the variance of these group means, weighted by the number of samples in each group (inter-group variation). Finally, the intra-group and inter-group variations are combined into a single stability value, which represents the stability of the gene combination. This process explains the observed difference between the stability of individual miRNAs and that of gene combinations. We have discussed this difference in Discussion section, from line 308 through 317, in the revised Word file on page 11.

Line 183:
“A one-way ANOVA with Tukey’s multiple comparisons test revealed no significant differences between the treated groups and the control group.” This result is not shown in the figure. Including this data would enhance the manuscript's transparency.

: As recommended, we have included the qPCR results for individual miRNAs in NE-4C cells as Figure 1 in the revised Word file on page 6.

  1. Reliability of Reference miRNAs:
    Line 221 states, “miR-181d-5p, miR-26a-5p, miR-125a-5p, and miR-361-5p were the most reliable reference miRNAs in human HP tissue.” The rationale behind this conclusion is unclear. miR-103-3p and let-7d-5p also appear to be reliable reference miRNAs in both mouse and human HP tissues, with let-7d-5p showing the lowest stability value (0.005).

: We completely agree that our previous description could cause confusion in understanding the results. Therefore, we have revised the description to clarify the results.

Figure 4:
Panels A to I are not labeled, which makes it difficult to interpret the data.

: This was our oversight, which occurred while transferring the figure from the PowerPoint file to the Word file. We have corrected this error in Figure 4 on page 9.

Round 2

Reviewer 2 Report

Comments and Suggestions for Authors

The authors have addressed my concerns.I have no further comments